# Universal mechanical exfoliation of large-area 2D crystals

Yuan Huang[1,2,12], Yu-Hao Pan[3,12], Rong Yang[1,2,12], Li-Hong Bao [1], Lei Meng[1], Hai-Lan Luo [1], Yong-Qing Cai[1], Guo-Dong Liu[1], Wen-Juan Zhao[1], Zhang Zhou[1], Liang-Mei Wu[1], Zhi-Li Zhu[1], Ming Huang [4], Li-Wei Liu[5], Lei Liu [6], Peng Cheng[1], Ke-Hui Wu[1], Shi-Bing Tian[1], Chang-Zhi Gu [1], You-Guo Shi[1], Yan-Feng Guo[7], Zhi Gang Cheng [1,2,8], Jiang-Ping Hu[1,2,8], Lin Zhao[1,2,8], Guan-Hua Yang[9], Eli Sutter [10], Peter Sutter [11✉], Ye-Liang Wang[1,4], Wei Ji [2✉], Xing-Jiang Zhou[1,2,8✉] & Hong-Jun Gao [1,8✉]

Two-dimensional materials provide extraordinary opportunities for exploring phenomena arising in atomically thin crystals. Beginning with the first isolation of graphene, mechanical exfoliation has been a key to provide high-quality two-dimensional materials, but despite improvements it is still limited in yield, lateral size and contamination. Here we introduce a contamination-free, one-step and universal Au-assisted mechanical exfoliation method and demonstrate its effectiveness by isolating 40 types of single-crystalline monolayers, including elemental two-dimensional crystals, metal-dichalcogenides, magnets and superconductors. Most of them are of millimeter-size and high-quality, as shown by transfer-free measurements of electron microscopy, photo spectroscopies and electrical transport. Large suspended two-dimensional crystals and heterojunctions were also prepared with high-yield. Enhanced adhesion between the crystals and the substrates enables such efficient exfoliation, for which we identify a gold-assisted exfoliation method that underpins a universal route for producing large-area monolayers and thus supports studies of fundamental properties and potential application of two-dimensional materials.

[1] Institute of Physics, Chinese Academy of Sciences, 100190 Beijing, China. [2] Songshan Lake Materials Laboratory, 523808 Dongguan, China. [3] Department of Physics and Beijing Key Laboratory of Optoelectronic Functional Materials & Micro-Nano Devices, Renmin University of China, 100872 Beijing, China. [4] School of Materials Science and Engineering, Ulsan National Institute of Science and Technology (UNIST), Ulsan 44919, Republic of Korea. [5] School of Information and Electronics, MIIT Key Laboratory for Low-Dimensional Quantum Structure and Devices, Beijing Institute of Technology, 100081 Beijing, China. [6] College of Engineering, Peking University, 100871 Beijing, China. [7] School of Physical Science and Technology, Shanghai Tech University, 201210 Shanghai, China. [8] University of Chinese Academy of Sciences, 100049 Beijing, China. [9] Institute of Microelectronics of Chinese Academy of Sciences, 100029 Beijing, China. [10] Department of Mechanical and Materials Engineering, University of Nebraska—Lincoln, Lincoln, NE 68588, United States. [11] Department of Electrical and Computer Engineering, University of Nebraska—Lincoln, Lincoln, NE 68588, United States. [12] These authors contributed equally: Yuan Huang, Yu-Hao Pan, Rong Yang. ✉email: psutter@unl.edu; wji@ruc.edu.cn; xjzhou@iphy.ac.cn; hjgao@iphy.ac.cn

Two-dimensional (2D) materials continue to reveal dimensionality-correlated quantum phenomena[1–6], such as 2D superconductivity, magnetism, topologically protected states, and quantum transport[1,7–11]. Stacking 2D materials into van der Waals heterostructures leads to further emergent phenomena and derived device concepts[12,13]. The further discovery and application of their properties depend on the development of synthesis strategies for 2D materials and heterostructures[6,14–18]. Synthesis using crystal growth methods can now produce large (millimeter scale) single crystals of some 2D materials, notably graphene and hexagonal boron nitride, but the scalable growth of high-quality crystals has remained challenging[18,19], with many 2D materials and especially heterostructures proving difficult to realize by bottom-up approaches. Ion-intercalation and liquid exfoliation are used as top-down approaches but, as chemical methods, they often cause contamination of the isolated 2D surfaces[20,21]. While some gold-assisted exfoliation methods were demonstrated in layered chalcogenides[22–24], those methods still bring unexpected contamination in samples prepared for electrical and optical measurements and thus reduce their performances when removing gold films with chemical solvents in additional steps. In light of this, a contamination-free, one-step and universal preparation strategy for large-area, high-quality monolayer materials is still lacking for both fundamental research and applications.

In the past 15 years, mechanical exfoliation has been a unique enabler of the exploration of emergent 2D materials. Most intrinsic properties of graphene, such as the quantum Hall effect[25], massless Dirac Fermions[26], and superconductivity[27], were mostly observed on exfoliated flakes but are either inaccessible or suppressed in samples prepared by other methods[17,28]. While exfoliation often suffers from low yield and small sizes of the exfoliated 2D flakes[5], many layered materials are, however, yet to be exfoliated into monolayers by established exfoliation methods. Such challenge of exfoliation limits their utility for scalable production of 2D crystals and complicates further processing, e.g., to fabricate heterostructures. These issues could be resolved by identifying suitable substrates that firmly adhere to 2D crystals without compromising their structure and properties, thus allowing the separation and transfer of the entire top sheet from a layered bulk crystal. Covalent-like quasi-bonding (CLQB), a recently uncovered non-covalent interaction with typical interaction energies of ~0.5 eV per unit cell[29–31], fits the requirements of the craving interaction between substrates and 2D layers. The intermediate interaction energy for CLQB is a balance of a reasonably large Pauli repulsion induced by interlayer wavefunction overlap and an enhanced dispersion attraction caused by more pronounced electron correlation in 2D layers with high polarizability.

Promising candidate substrates for CLQB with 2D crystals are materials whose Fermi level falls in a partially filled band with mostly s- or p-electrons to prevent disrupting the electronic structure of 2D layers, and which have highly polarizable electron densities to ensure a large dispersion attraction. Noble metals meet these criteria and are easily obtained as clean solid surfaces. Group 11 (IB) coinage metals, i.e., Cu, Ag, and Au, remain as potential candidates after ruling out group 8–10 (VIII) metals of too strong hybridization, Al of high activity to 2D layers and in air[32,33] and closed-shell group 12 (IIB) metals (Zn, Cd, Hg). Among those three, Au interacts strongly with group 16 (VIA) chalcogens (S, Se, Te) and 17 (VIIA) halogens (Cl, Br, I), which terminate surfaces in most 2D materials. Together with its low chemical reactivity and air stability, Au appears promising for high-yield exfoliation of many 2D materials, which is also evidenced by three previous reports[22–24]. According to the periodic table, Pt may behave similarly to Au. However, the hybridization between Pt and many 2D appears too strong to significantly change their electronic structures[34]. In addition, Au is mechanically softer than Pt, which may improve the interfacial contact under the gentle pressure applied during the exfoliation process.

## Results

**Theoretical prediction of Au-assisted exfoliation**. Density functional theory (DFT) calculations were employed to substantiate these arguments by comparing the interlayer binding energies of a large set of layered crystals with their adhesion energies to the Au (111) surface. A total of 58 layered materials, including 4 non-metallic elemental layers and 54 compounds comprised of metal and non-metal elements were considered in our calculations (see Fig. 1a). They belong to 18 space groups covering square, hexagonal, rectangular, and other lattices (Fig. 1b). The surfaces of all considered compound layers are usually terminated with group 16 (VIA) or 17 (VIIA) elements, e.g., S, Se, Te, Cl, Br, and I, with the exception of $W_2N_3$. These atoms, together with group 15 (VA) elements, are expected to have substantial interactions with Au substrates, which is verified by our differential charge density (DCD) plots.

Figure 1c–f shows the DCDs of graphene, black phosphorus (BP), $MoS_2$, and $RuCl_3$ monolayers adsorbed on Au (111), representing the interactions of Au with group 14 (IVA) to 17 (VIIA) atoms, respectively. The adhesion induced charge redistribution of graphene differs from those of the other three layers. While Au only introduces charge dipoles at the interface to graphene, significant covalent characteristics, i.e., charge reduction near the interfacial atoms and charge accumulation between them, were observable at the P/Au, S/Au, and Cl/Au interfaces. The difference in charge redistribution is reflected in the smaller adhesion energy of graphene/Au (28 meV Å$^{-2}$; 0.15 eV per unit cell) compared with those of the other three interfaces (56, 40, and 36 meV Å$^{-2}$; 0.80, 0.35, and 1.11 eV per unit cell). The clearly covalent nature of the S/Au interface is consistent with previous reports[22–24] and confirms our expectation. Our results of DCD and electronic band structures (Supplementary Fig. 1), suggest the existence of CLQB at the S/Au, P/Au, and Cl/Au interfaces, which is confirmed by comparing the interlayer (0.23, 0.48, and 0.57 eV per unit cell) and 2D crystal/Au (0.35, 0.80, and 1.11 eV per unit cell) binding energies. Figure 1g and Supplementary Table 1 show the comparison of these energies for all 58 considered 2D crystals, where the 2D crystal/Au binding is invariably stronger than the corresponding interlayer binding. These results support the concept that the 2D crystal/Au interaction should be sufficient to overcome the interlayer attraction and facilitate exfoliating monolayers from a broad range of layered crystals. Here, we define a ratio $R_{LA/IL}$ as layer-Au over interlayer adhesion energies. Possible exceptions are those 2D materials whose $R_{LA/IL}$ values, while greater than 1, are substantially smaller than usual $R_{LA/IL}$ values (>1.3). Here, BN (1.07), $GeS_2$ (1.17), and graphene (1.24) are some examples.

**Large-scale exfoliation of 40 two-dimensional monolayers**. To test these theoretical predictions, we implemented the Au-assisted exfoliation of 2D materials as shown in Fig. 2a. Firstly, a thin layer of Au is deposited onto a substrate covered with a thin Ti or Cr adhesion layer. Then, a freshly cleaved layered bulk crystal on tape is brought in contact with the Au layer. Adhesive tape is placed on the outward side of the crystal, and gentle pressure is applied to establish a good layered crystal/Au contact. Peeling off the tape removes the major portion of the crystal, leaving one or few large-area monolayer flakes on the Au surface. Limited only by the size of available bulk crystals, these monolayer flakes are usually macroscopic in size (millimeters; see Methods for details).

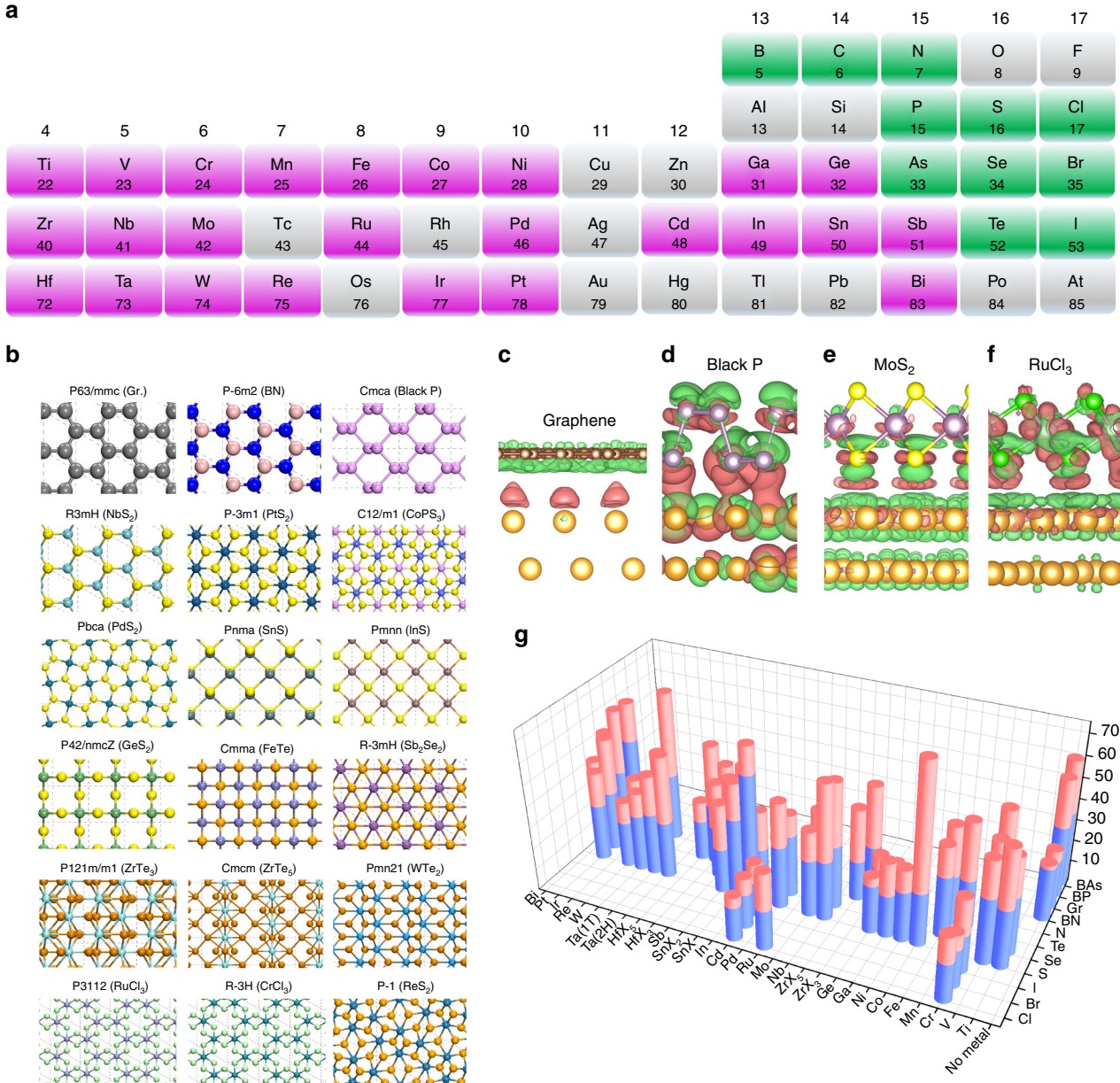

**Fig. 1 DFT calculated interlayer binding energies of 2D materials and adsorption energies on Au (111) surfaces. a** Part of the periodic table, showing the elements involved in most 2D materials between groups 4 (IVB) and 17 (VIIA). **b** Eighteen space groups and typical structural configurations (top views) of the 2D materials. **c–f** DCD of four Au (111)/2D crystal interfaces with (non-metallic) terminating atoms between groups 14 (IVA) and 17 (VIIA). Isosurface values of these DCD plots are $5 \times 10^{-4}$ e Bohr$^{-3}$ (graphene), $1 \times 10^{-2}$ e Bohr$^{-3}$ (BP), and $1 \times 10^{-3}$ e Bohr$^{-3}$ (MoS$_2$, RuCl$_3$), respectively. **g** Bar graph comparing the interlayer binding energies of 2D materials (blue cylinders) with their adsorption energies on Au (111) (red cylinders). The visible red cylinders represent the difference between the Au/2D crystal interaction and the interlayer interaction.

Optical microscopy was used to examine the dimensions and uniformity of the exfoliated 2D crystals. Figure 2b shows an image of exfoliated MoS$_2$ monolayers reaching lateral dimensions close to 1 cm on a SiO$_2$/Si substrate covered with Au (2 nm)/Ti (2 nm). We also extended the base substrate from the SiO$_2$/Si substrate to transparent (quartz, sapphire; Fig. 2c) and flexible plastic supports (Fig. 2d). The transparency persists even for thicker (~10 nm) Au/Ti layers although light transmission slightly decreases. This method can also be applied to CVD-grown wafer-scale transition-metal-dichalcogenides (TMDCs) materials, such as MoS$_2$ (Fig. 2e). The exfoliated monolayer flakes can be intactly transferred onto arbitrary substrates after removing gold layer by KI/I$_2$ etchant. Therefore, back-gated devices and heterostructures can be fabricated (see Methods and Supplementary Fig. 5 for details).

X-ray photoelectron spectroscopy (XPS) was employed to further investigate the interaction between MoS$_2$ and Au. Supplementary Fig. 2d shows an XPS spectrum of exfoliated MoS$_2$ near the Mo 3d region. Peaks centered at 226.5, 229, and 232 eV result from S 2 s, Mo 3$d_{5/2}$, and Mo 3$d_{3/2}$ photoelectrons, respectively. There are no appreciable changes in terms of shape, binding energy, and width of the XPS peaks compared to those of bulk MoS$_2$. Hence, the nearly unchanged XPS spectra confirm CLQB rather than covalent bonding between MoS$_2$ and the Au substrate. Supplementary Figs. 3 and 7 show Raman, photoluminescence (PL) and angle-resolved photoemission spectroscopy (ARPES) of a typical exfoliated MoS$_2$ monolayer. Sharp $E_{2g}$ and $A_{1g}$ Raman peaks at 386 and 406 cm$^{-1}$, respectively, confirm the high quality of the MoS$_2$ monolayer[35]. The pronounced A-

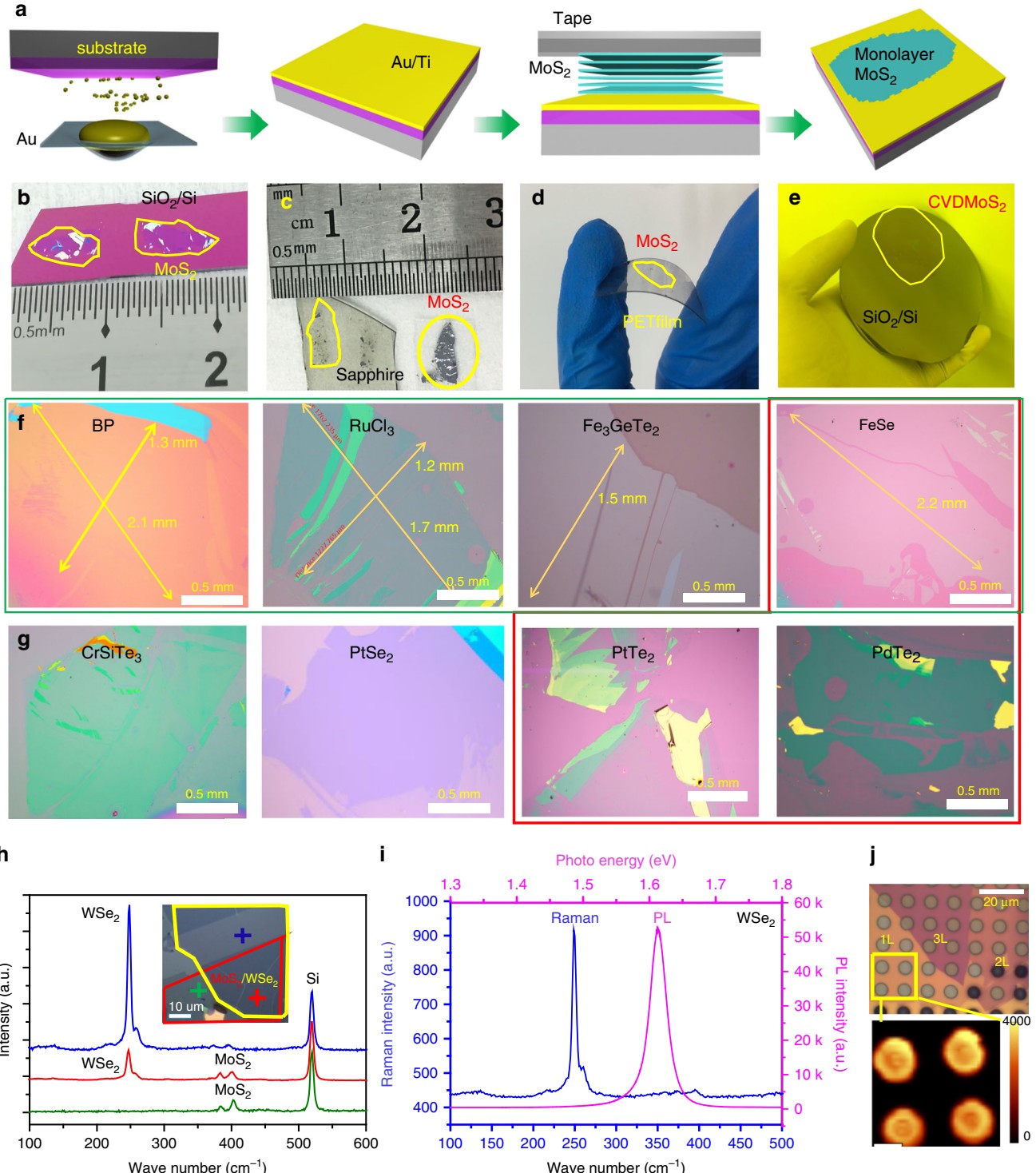

**Fig. 2 Mechanical exfoliation of different monolayer materials with macroscopic size. a** Schematic of the exfoliation process. **b–d** Optical images of exfoliated MoS$_2$ on SiO$_2$/Si, sapphire, and plastic film. **e** 2-inch CVD-grown monolayer MoS$_2$ film transferred onto a 4 inch SiO$_2$/Si substrate. **f–g** Optical images of large exfoliated 2D crystals: BP, FeSe, Fe$_3$GeTe$_2$, RuCl$_3$, PtSe$_2$, PtTe$_2$, PdTe$_2$, and CrSiTe$_3$. Those exfoliated monolayers highlighted in the red box are, so far, not accessible using other mechanical exfoliate method. **h** Optical image and Raman spectra of a MoS$_2$/WSe$_2$ heterostructure. **i** Raman and photoluminescence (PL) spectra of suspended monolayer WSe$_2$. **j** Optical image of suspended WSe$_2$ with different thicknesses (1 L to 3 L) and a PL intensity map of the suspended monolayer.

exciton PL peak at 1.83 eV indicates the exfoliated MoS$_2$ on Au is still a direct-band gap semiconductor[36]. Metal substrates usually quench the PL intensity of monolayer MoS$_2$[23,24]; however, our PL signal remains strong because of the thickness-tunable conductivity of our metal substrates, as elucidated later.

We applied the Au-assisted exfoliation method to other 2D crystals and have obtained a library of 40 large-area single-crystal monolayers, as shown in Fig. 2f–g and Supplementary Fig. 8. Besides transition-metal-dichalcogenides, the library contains metal monochalcogenides (e.g., GaS), black phosphorus, black

arsenic, metal trichlorides (e.g., $RuCl_3$), and magnetic compounds (e.g., $Fe_3GeTe_2$). It is rather striking that some monolayers, i.e., FeSe, $PdTe_2$, and $PtTe_2$, become accessible by our exfoliation method. This method is, as we expected according to the smaller $R_{LA/IL}$ values (1.24 and 1.07), less effective for exfoliating graphene and h-BN monolayers, which are accessible by chemical vapor deposition. The exfoliated monolayer samples show high quality, as characterized by Raman and atomic force microscopy (Supplementary Figs. 9 and 10). Reactive samples were exfoliated in a glove box due to their stability issues in air.

**Optical characterization of hetero- and suspended-structures.** Our method also promotes preparation of van der Waals heterostructures and suspended 2D materials at human visible size scales. Figure 2h shows a typical monolayer $MoS_2/WSe_2$ heterostructure prepared using this method. Raman spectra (Fig. 2h) show the characteristic vibrational modes of both the $MoS_2$ and $WSe_2$ layers. PL spectra of this heterostructure sample is shown in Supplementary Fig. 6. Given the exceeding $R_{LA/IL}$ values over 1.30, patterned Au thin-films on substrates with holes, are also, most likely, able to exfoliate 2D crystals and thus to fabricate suspended monolayers, which is of paramount importance on studying intrinsic properties of 2D layers[37,38]. We show an example with suspended 1L-3L $WSe_2$ in (Fig. 2i), which can reach 90% coverage over at least tens of micrometers (Supplementary Fig. 11). The suspended monolayer film is detached from multilayer instead of transferring monolayer by organic films, which totally avoid polymer contamination. In comparison with supported samples on $SiO_2$, the suspended $WSe_2$ (Fig. 2i) shows enhanced PL intensity (16 times) and sharper PL peak (full width at half maximum (FWHM): 34 meV, compared with 64 meV for supported $WSe_2$) as shown in Supplementary Fig. 12. Since PL can be fully quenched on thicker metal film while well maintained on suspended area, therefore, we realized patterning of PL even on one monolayer flake (Fig. 2j).

**Surface characterization of as-exfoliated samples.** High-quality macroscopic monolayers have practical advantages, for instance in establishing the lattice structure and electronic band structure of unexplored 2D materials or van der Waals stacks by scanning tunneling microscopy (STM) and ARPES. The Au-coated support facilitates such electron-based spectroscopy by eliminating charging effects associated with insulating (e.g., $SiO_2$) substrates while preserving the intrinsic electronic band structures. Figure 3a, b illustrates atomic-resolution STM images for as-exfoliated $WSe_2$ and $T_d$-$MoTe_2$ monolayers, which are challenging to image on insulating substrates due to charging effects. Low-energy electron diffraction with millimeter incident electron-beam size shows a single-phase diffraction pattern for $MoTe_2$ (Fig. 3c), indicating that it is a single-crystal at the millimeter scale. Figure 3d displays an ARPES map of the low-energy electronic structure of the $WSe_2$ monolayer, showing clear and sharp bands. The valence band features a single flat band around $\Gamma$ and a large band splitting near K. Along the $\Gamma$-K line, one single band starts to split into two spin-resolved bands at $k \approx \frac{1}{3}\Delta k^{K,\Gamma}$, and the valence band maximum at K sits at ~0.6 eV higher than that at $\Gamma$. Figure 3e displays the symmetric band splitting spectra along K–M–K' arising from strong spin-orbit coupling mainly at the W site in the $WSe_2$ lattice[39]. These features constitute the critical signatures of band dispersion in monolayer TMDCs. Here it deserves an emphasis on the big advantage of large area of monolayer TMDCs, which make it quite feasible and easy to accurately measure the band structure by using standard ARPES technique[40].

**FET Devices directly built on as-exfoliated layers.** Even some previous studies proved to exfoliate large-area chalcogenides layers using gold films[22–24]; however, further characterizations including optical and electrical measurements are usually achieved by an additional transfer process onto insulating substrate. Nevertheless, we show here that the conductivity of Au/Ti adhesion layer can be drastically tuned by controlling its nominal thickness, which demonstrate that both optical measurements and device fabrication can be realized on this one-step exfoliated samples. Figure 4a, b shows that the Au/Ti films become insulating (i.e., electrically discontinuous) if the combined thickness of Ti/Au decreases to 3 nm or below[41]. Our exfoliation method is not apparently limited by the Au thickness, therefore, those large-area 2D crystals are expected to be exfoliated by either conducting or non-conducting Au-coated substrates.

Supplementary Fig. 3c shows a $MoS_2$ flake exfoliated by an electrically discontinuous (0.5 nm Ti, 1.5 nm Au) adhesion layer on a $SiO_2/Si$ substrate (Fig. 4b inset and Supplementary Fig. 13), in which almost no lateral channel is available to carry current flow through the substrate. Although some prior studies used Au films to enhance exfoliation of $MoS_2$, the PL intensity of their $MoS_2$ samples was largely quenched[22,24]. By contrast, our $MoS_2$ monolayer exfoliated using electrically discontinuous metal adhesion layers show intense PL signals (see Supplementary Figs. 3 and 4). Such electrically discontinuous layers also allow to fabrication of electronic devices directly from the as-exfoliated 2D monolayers. Figure 4c shows the trans-conductance curve of a prototype device, a field-effect transistor (FET) directly built on an as-exfoliated monolayer $MoS_2$ channel on Au(1.5 nm)/Ti (0.5 nm)/$SiO_2$/Si. Supplementary Fig. 14 shows the device layout. The device, controlled by an ionic-liquid top gate, shows a high on–off current ratio (>$10^6$ at $T = 220$ K), comparable to usual $MoS_2$ FETs directly fabricated on $SiO_2$/Si substrates[7,42]. A sub-threshold swing (SS) of 100 mV dec$^{-1}$. was derived, close to the best values reported in the literature[7,43,44], ranging from 74 mV dec$^{-1}$. to 410 mV dec$^{-1}$. Given the previously reported capacitance value of $C_i = 1.3$~2 $\mu$F cm$^{-2}$ for the ionic liquid[45,46], we derived the field-effect mobility of 22.1–32.7 cm$^2$V$^{-1}$s$^{-1}$ for our ionic-liquid-gated $MoS_2$ device (Fig. 4c). The estimated mobility value is very close to that of our ordinary back-gated FET built with transferred $MoS_2$ monolayer on $SiO_2$/Si (~30 cm$^2$V$^{-1}$s$^{-1}$) and is also comparable with those values reported in the literature, i.e., tens of cm$^2$V$^{-1}$s$^{-1}$ [47]. A superconducting transition was also observed at 4.5 K (Supplementary Fig. 15). All these results show good performance of the FET directly fabricated on the ultrathin metal adhesion layer and its potential for further improvements. We extended the individual FET to an FET array directly built on an exfoliated centimeter-scale single-crystal $MoS_2$ flake using UV lithography (Supplementary Fig. 16), indicating great potential of our exfoliation method for fabrication of integrated circuit. Note that, for those FET devices with discontinuous metal layer underneath, the back gate does not work due to the screen effect. However, the 2D layers could be lifted off and transferred to other insulating substrates, e.g., oxidized Si wafer, by removing the gold layer. We also fabricated back-gated $MoS_2$ monolayer FET devices based on freshly transferred large-size $MoS_2$ flakes on $SiO_2$/Si substrates. Supplementary Fig. 17 presents the results of electrical transport measurements of the back-gated FETs, which show high mobility (~30 cm$^2$V$^{-1}$s$^{-1}$) and large on–off ratio (~$10^7$). These values manifest that the transferred $MoS_2$ monolayer still preserves its high quality (Supplementary Fig. 17; see Methods for details).

Since electrical current is prone to flow through superconducting regions, the metal adhesion layer does not suppress the transition of an exfoliated layer from its normal state to its

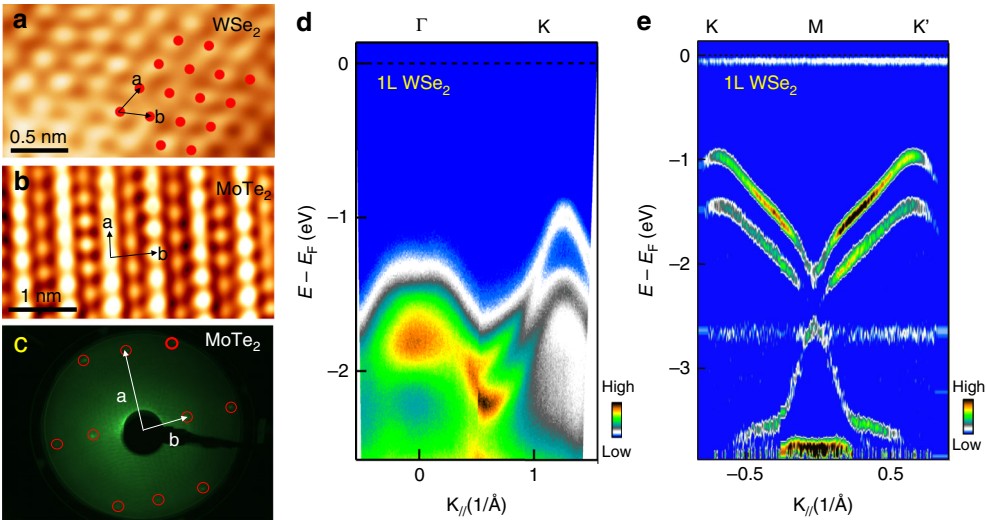

**Fig. 3 STM and ARPES measurements of 2D materials exfoliated onto conductive Au/Ti adhesion layers. a, b** STM images of monolayer WSe$_2$ and T$_d$-MoTe$_2$, respectively. **c** LEED pattern of monolayer T$_d$-MoTe$_2$. **d, e** Band structure of monolayer WSe$_2$. **d** Original ARPES band structure of monolayer WSe$_2$ ($hv = 21.2$ eV) along Γ-K high symmetry line. The valence band maximum (VBM) is positioned at K instead of Γ, which is an important signature of monolayer WSe$_2$. **e** Second-derivative spectra of band dispersion along K–M–K′, showing clear spin-orbital coupling (SOC) induced spin-splitting bands.

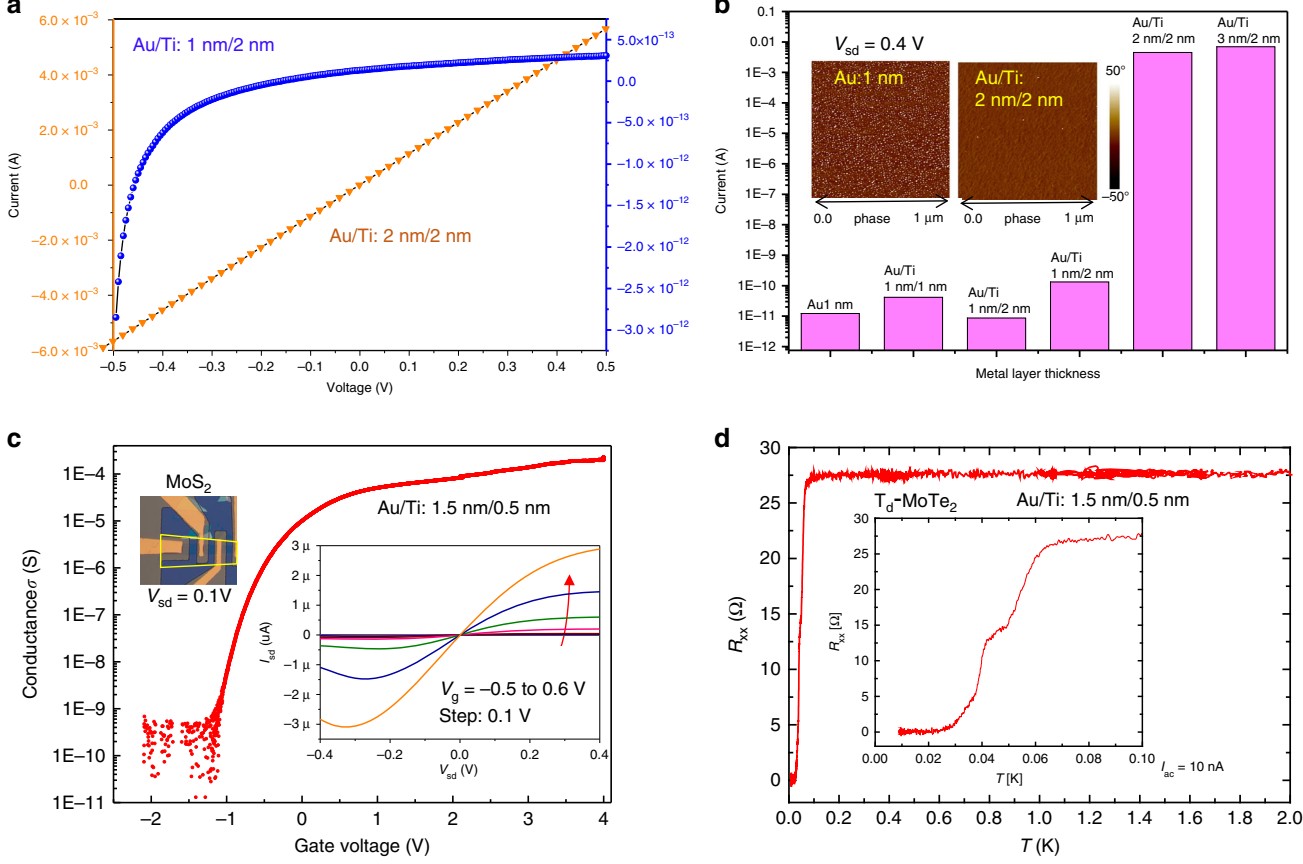

**Fig. 4 Electrical measurements of metal adhesion layers and of 2D materials exfoliated onto nonconductive metal films. a** Electrical transfer curves of typical Au/Ti adhesion layers. **b** Two-terminal resistance of Au/Ti layers with different nominal thickness. The inset shows atomic force microscope (AFM) phase maps of two metal layers. **c** Gate voltage-conductance transfer characteristics of a top-gated MoS$_2$ FET on SiO$_2$/Si with Au (1.5 nm)/Ti (0.5 nm) adhesion layer ($T = 220$ K, source−drain bias $V_{sd} = 0.1$ V). Left inset: Optical image of the FET device with windows for the ionic-liquid top gate. Right inset: low-bias source-drain current-voltage characteristics for gate voltage −0.5 to 0.6 V. **d** Temperature-dependent resistance of a T$_d$-MoTe$_2$ flake exfoliated onto SiO$_2$/Si with a 2 nm metal adhesion layer.

superconducting state. Figure 4d shows a $T_d$-$MoTe_2$ device, directly fabricated on a 2 nm metal layer, undergoes a metal-superconductor transition at 70 mK, and reaches the zero-resistance at 30 mK. A second onset, observed at 50 mK, is, most likely, a result of quantum fluctuation in 2D crystals. A previous study of bulk $T_d$-$MoTe_2$ showed an onset of superconductivity at 250 mK and zero-resistance at a critical temperature $T_c = 100$ mK[48]. The difference between bulk and monolayer's $T_c$ values may be primarily relevant with reduced dimensionality[49].

## Discussion

Our combined results show that exfoliation assisted by an Au adhesion layer with covalent-like quasi bonding to a layered crystal provides access to a broad spectrum of large-area mono-layer materials. This method is rather unique, especially for layered crystals that are difficult to exfoliate using conventional methods. The versatility of this approach is demonstrated here by using Au adhesion layers for exfoliation of large 2D sheets from 40 layered materials. The efficient transfer of most 2D crystals is rationalized by calculations that indicate interaction energies to Au exceeding the interlayer energy for most layered bulk crystals, graphene and hexagonal boron nitride being notable exceptions. Characterization of the large-area exfoliated monolayers flakes demonstrates that the flakes are of high quality. For research on atomically thin materials, the approach demonstrated here has immediate implications. The availability of macroscopic (milli-meter scale) 2D materials can support the exploration of the properties of emergent families of ultrathin semiconductors, metals, superconductors, topological insulators, ferroelectrics, etc., as well as engineered van der Waals heterostructures. For applications of 2D materials, an efficient large-scale layer transfer method could force a paradigm shift. So far, exfoliation from bulk crystals has not been deemed technologically scalable. But once exfoliation becomes so consistent that the size of the resulting 2D layers is limited only by the dimensions and crystallinity of the source crystal, the focus of application-driven materials research may shift toward optimizing the growth of high-quality layered bulk crystals. Ironically, the fabrication of 2D materials for applications would then follow the well-established and highly successful example of silicon technology, where the extraction of wafers from large, high-quality single crystals has long been key to achieving the yields and reliability required for industrial applications.

## Methods

**DFT calculations**. DFT calculations were performed using the generalized gradient approximation for the exchange-correlation potential, the projector augmented wave method[50,51], and a plane-wave basis set as implemented in the Vienna ab initio simulation package (VASP)[52]. The energy cutoff for the plane-wave basis set was set to 700 and 500 eV for variable volume structural relaxation of pure 2D materials and invariant volume structural relaxation of these materials on Au (111) surface, respectively. Dispersion correction was made at the van der Waals density functional (vdW-DF) level, with the optB86b functional for the exchange potential[53]. Seven Au (111) layers, separated by a 15 Å vacuum layer, were employed to model the surface. The four bottom layers were kept fixed and all other atoms were fully relaxed until the residual force per atom was less than 0.04 eV Å$^{-1}$ during structural relaxations of 2D layers on Au (111) and less than 0.01 eV Å$^{-1}$ during all other structural relaxations. Lattice constant of the Au (111) surface slab was varied to match those of 2D layers for keeping their electronic properties unchanged by external strain. The lattice mismatch between each 2D layer and the Au (111) surface was kept lower than 4.5%. A $k$-mesh of $9 \times 9 \times 1$ was adopted to sample the first Brillouin zone of the primitive cell of the Au(111) slab and the density of $k$-mesh was kept fixed in other relaxation calculations. Differential charge density (DCD) between a 2D layer and the Au substrate represents the charge variation after the 2D layer attaching to Au and was derived using $\Delta\rho_{DCD} = \rho_{All} - \rho_{Au} - \rho_{2D}$. Here $\rho_{All}$ is the total charge density of the 2D layer/Au(111) interface, while $\rho_{Au}$ and $\rho_{2D}$ are the total charge densities of the individual Au surface and the 2D layer, respectively.

**Gold-assisted mechanical exfoliation**. The metal layer deposition was completed in an electron evaporation system (Peva-600E). An adhesion metal layer (Ti or Cr) was first evaporated on Si substrate (with 300 nm $SiO_2$ film), after that Au film was deposited on the substrate. The thickness of Ti (or Cr) and Au can be well controlled by the evaporation rate (0.5 Å s$^{-1}$). After depositing metal layers on Si wafer, a fresh surface of layered crystal was cleaved from tape and put it onto the substrate. By pressing the tape vertically for about 1 min, the tape can be removed from substrate. Large-area monolayer flakes can be easily observed by optical microscope or even by eyes. Most of the time, the size of monolayer flakes is limited by the size of bulk crystal. The glue residues mainly depend on different tapes, for the white tape (3 M scotch) or blue tape (Nitto), there will be some glue left on substrate, which can be removed by further annealing if required by the subsequent measurement. However, we can also use home-made poly-dimethylsiloxane (PDMS) to replace these tapes as transfer media, which can make the substrate much cleaner[54]. The thickness of Au/Ti layer is a crucial factor. Once the thickness larger than 2 nm Au/ 1 nm Ti, the success rate is more than 99.5%. However, the flake size will decrease to few hundred micrometers when the metal layer thinner than 1 nm Au/ 1 nm Ti. It is hard to get 2D flakes when metal thickness thinner than 0.5 nm Au/ 0.5 nm Ti. The optimal thickness of Au/Ti is 2 nm/ 2 nm. Therefore, when the gold film is nonconductive and rough, the success rate will be affected. In terms of the substrate with hole array, the success rate for exfoliating suspended $MoS_2$ is also more than 99% once the metal thickness is larger than Au(2 nm)/Ti(1 nm) and diameter of hole smaller than 5 μm. The success rate decreases obviously if the hole diameter larger than 10 μm.

The bulk crystals are mainly supplied by Dr. You Guo Shi's and Dr. Yanfeng Guo's groups. We also tested some crystals (such as $MoS_2$, $WSe_2$) from commercial companies, like HQ Graphene and 2D Semiconductors, all these crystals can be exfoliated into large-area monolayer, and the monolayer size mainly depends on the bulk crystal. The thickness of bulk crystal has some influence for the exfoliation process, but not very crucial. Once the bulk crystal is too thick (e.g., >1 mm) it will be difficult to ensure the interface between crystal and substrate contact well. However, it is not necessary to make the bulk crystals too thin on tape (e.g., monolayer or few layer), because bulk crystal can be break into small pieces if cleave too many times by tape. Normally, we cleave 2–3 times from pristine bulk crystal before put onto substrate with metal film. The temperature and humidity is not very sensitive for the exfoliation, the authors and some of collaborators tested this exfoliation method in different counties (US, China, Singapore and Korea) and in different seasons (the whole year from Spring to Winter), we can always get large-area 2D flakes with high success rate. The exfoliation processes of the 2D materials mainly carried out in clean-room, and the temperature and humidity always keep at 25 °C and 45%. While the exfoliation can be done in a glove box without exposure to air, the whole process lasts 1–2 min.

**Suspended samples preparation**. Si wafer with 300 nm $SiO_2$ was patterned by UV lithography, after that the hole array structures were prepared by reactive-ion etching. The diameter and depth of each hole is 5 μm and 10 μm. The metal layers (Au/Ti: 2 nm/2 nm) deposited on the Si substrate with hole array before exfoliating layered materials on it. Large-area suspended 2D materials can be exfoliated on the hole array substrate.

**Heterostructure preparation**. Schematic images for the details of fabrication procedure were shown in Supplementary Fig. 5. Firstly, we exfoliated large-area $MoS_2$ and $WSe_2$ monolayers on two separate Au/Ti/$SiO_2$/Si substrates, respectively. Then, PMMA was spin-coated onto the $MoS_2$ monolayer and then the sample was put into KI/$I_2$ solution. After roughly 10 h of etching, the gold film was removed and the PMMA film together with the $MoS_2$ flake detached from the $SiO_2$/Si substrate. In order to clean the ion residual, the PMMA film was washed three times using DI water. The next step lies in using the $WSe_2$ sample to pick up the PMMA/$MoS_2$ film from water. Since both $MoS_2$ and $WSe_2$ flakes are several millimeters in size, no special alignment is need in this step if the twisting angle between these two layers is not specified. Additional baking at ~100 °C ensures the contact between these two layers. Next, we employed the same procedure, i.e., etching in KI/$I_2$ solution and three times washing using DI water, to remove the Au film from $WSe_2$. A new substrate, e.g., $SiO_2$/Si was used to pick up the hetero-bilayer from Di water. Finally, we removed the PMMA layer using acetone.

**Monolayer $MoS_2$ back-gated FET devices fabrication**. After exfoliating large-area $MoS_2$ on metal film (Au/Ti: 2 nm/2 nm), PMMA films were spin-coated on the substrates. Then, the samples were put into KI/$I_2$ solution for about 10 h. The PMMA films together with $MoS_2$ can be detached from $SiO_2$/Si substrate after etching gold film. The PMMA films with $MoS_2$ were cleaned three times by DI water, after that the films were transferred onto new Si substrates (with 300 nm $SiO_2$ layer) for device fabrication. Electron-beam lithography (EBL) is used to pattern an etch mask using poly(methyl) methacrylate (PMMA). The devices were fabricated after metal deposition (50 nm Au, 10 nm Ti) and lift-off.

**Optical characterization and measurement**. The Raman and PL measurements were performed on WITec alpha300R and JY Horiba HR800 systems with a wavelength of 532 nm and power at 0.6 mW. Supplementary Fig. 9 presents the

representative Raman spectra for monolayer and few-layer BP and α-RuCl₃ samples excited by 2.33 eV radiation in vacuum environments. The laser power on the sample during Raman measurement was kept below 100 μW in order to avoid sample damage and excessive heating. The silicon Raman mode at 520.7 cm$^{-1}$ was used for calibration prior to measurements and as an internal frequency reference.

**X-ray photoelectron spectroscopy characterization**. The X-ray photoelectron spectroscopy (XPS, Thermo Scientific ESCALAB 250 Xi) was performed with Al Kα X-rays (hν = 1486.6 eV) in an analysis chamber that had a base pressure < 3 × 10$^{-9}$ Torr. Core spectra were recorded using a 50 eV constant pass energy (PE) in 50–100 μm small area lens mode (i.e., aperture selected area). The XPS peaks were calibrated using the adventitious carbon C1s peak position (284.8 eV).

**Scanning probe microscopy measurements**. The AFM scanning (Veeco Multimode III) was used to check the thickness and surface morphology of those monolayer samples. The STM measurement was performed using a custom built, low-temperature, and UHV STM system at 300 K. A chemically etched W STM tip was cleaned and calibrated against a gold (111) single crystal prior to the measurements. For STM and ARPES measurements, the 2D layers were exfoliated onto an Au(5 nm)/Ti(2 nm)/SiO₂(~300 nm)/Si substrate. An annealing process at 500 K for 2 h was performed to degas the samples after loading them into the high-vacuum chamber.

**Angle-resolved photoemission spectroscopy measurement**. High resolution ARPES measurements were carried out on our lab system equipped with a Scienta R4000 electron energy analyzer[55]. We use Helium discharge lamp as the light source, which can provide photon energies of hν = 21.218 eV (Helium I). The energy resolution was set at 10–20 meV for band structure measurements (Fig. 3). The angular resolution is ~0.3 degree. The Fermi level is referenced by measuring on a clean polycrystalline gold that is electrically connected to the sample. The samples were measured in vacuum with a base pressure better than 5 × 10$^{-11}$ Torr. The ARPES measurements for WSe₂ and MoS₂ monolayers were carried out at roughly 30 K using a home-build photoemission spectroscopy system with a VUV5000 Helium lamp. The spot diameter of the Helium lamp is 0.5 mm.

**FET characterization and measurement**. The electrical characteristic measurements were carried out in the probe station with the semiconductor parameter analyzers (Agilent 4156 C and B1500) and oscilloscope. The ionic liquid used for top-gated MoS₂ FET device is N-diethyl-N-(2-methoxyethyl)-N-methylammonium bis-(trifuoromethylsulfonyl)-imide (DEME–TFSI), which has been widely used in 2D-material-based devices.

**Online content**. Methods, along with any Supplementary Information display items and Source Data, are available in the online version of the paper; references unique to these sections appear only in the online paper.

## Data availability

All data needed to evaluate the conclusions in the paper are present in the paper and/or the Supplementary Information. Additional data related to this paper may be requested from the authors.

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

## Acknowledgements
We would like to thank Dr. Zhongming Wei, Xianjue Chen and Jia Wang for valuable discussions about XPS. This work is supported by the National Key Research and Development Program of China (Grant No. 2019YFA0308000, 2018YFA0305800, 2018YFE0202700, 2018YFA0704201), the Youth Innovation Promotion Association of CAS (2019007, 2018013, 2017013), the National Natural Science Foundation of China (Grant No. 11874405, 11622437, 61674171, 61725107, 61971035, and 11974422), the National Basic Research Program of China (Grant No. 2015CB921300), and the Strategic Priority Research Program (B) of the Chinese Academy of Sciences (Grant No. XDB25000000, XDB30000000), the Research Program of Beijing Academy of Quantum Information Sciences (Grant No. Y18G06). P.S and E.S. acknowledge support by the U.S. Department of Energy, Office of Science, Basic Energy Sciences, under Award No. DE-SC0016343. Calculations were performed at the Physics Lab of High-Performance Computing of Renmin University of China and Shanghai Supercomputer Center.

## Author contributions
P.S., W.J., H.J.G., and X.J.Z. are equally responsible for supervising the discovery. Y.H. and R.Y. conceived the project. Y.H.P., J.P.H., and J.W. performed the DFT calculations. Y.H., H.L.L., and L.L. prepared all the mechanical exfoliation samples. Y.H. and R.Y. performed the Raman, PL and AFM measurements. M.H., J.W., P.S., and E.S. performed XPS measurement. Y.Q.C., G.D.L., L.Z., and W.J.Z. performed the ARPES measurement. Z.L.Z., P.C., K.H.W., L.M., Z.Z., L.W.L, and Y.L.W performed the STM and LEED measurement. Y.G.S. and Y.F.G. prepared bulk layered crystals. S.B.T., C.Z.G., Z.G.C., L.M.W. G.H.Y., and L.H.B. fabricated the transistors and performed the electrical measurements. Y.H., Y.H.P., R.Y., J.P.H., P.S., and W.J. analyzed data, wrote the manuscript and all authors discussed and commented on it.

## Competing interests
The authors declare the following competing interests that three Chinese patents were filed (201910529797.7; 201910529796.2; 201910529623.0) by the Institute of Physics, Chinese Academy of Sciences, along with their researchers (Y.H., H.L.L., and X.J.Z).
