## [Peer Review File · Nature Communications]

Reviewers' comments:

Reviewer #1 (Remarks to the Author):

Huang et al report a millimeter-scale 2D material exfoliation method assisted by the thin Au films deposited on the substrates. Au-assisted exfoliation was reported before to produce millimeter-scale transition-metal dichalcogenide (TMDC) monolayers. However, in this work, the authors have successfully extended the method to exfoliate up to 40 types of van der Waals materials, including semiconductors, magnets, and superconductors. It was not expected that Au-assisted exfoliation can work so well to different van der Waals materials other than TMDCs. The authors further demonstrated that their samples are suitable for angle-resolved photoemission spectroscopy (ARPES), scanning tunneling microscopy (STM), transport and optical measurements, which is not trivial since ARPES and STM are very sensitive to the sample quality and surface contaminations. These extensions certainly make the paper of interest to broader audiences. During the review process, the referee noticed another Au-assisted exfoliation paper published online on Science (Liu et al., Science 367, 903–906 (2020), online at 21.02), which demonstrates even larger exfoliated monolayers but still limited to TMDCs. The referee is convinced that this manuscript would be a very good complementary paper to the Science one. Thus, I would highly recommend the paper to be published on Nature Communications.

Nevertheless, there are still several issues to be addressed before the publication.

The major ones are:

1. The charge density plot strongly depends on the electronic state chosen. What are the states used for charge density plots in Fig. 1c-f? How were these states chosen and are they related to the Fermi level?
2. The data from STM and ARPES measurements in Fig. 3 looks very decent. But it is not clear how the samples were processed. What is the thickness of Au and Ti for the STM and ARPES samples? In general, monolayers for STM and ARPES characterization need to be annealed at a high temperature for hours to remove surface contaminations. Was the annealing process needed/performed for the authors' samples before the characterization? What was the size of the beam used for ARPES?
3. The authors claimed that the PL of their TMDC monolayers is not strongly quenched (Line 271) because of the thin layer of gold used. However, the PL spectrum of monolayer MoS₂ in Fig. S3b doesn't look very strong with the current signal to noise ratio. The scale bar for the PL imaging in Fig. S3d is missing. Figure S9b indicates that more than 10 times PL quenching occurs when there is Au underneath. To justify the claim, the authors should compare the monolayers produced by Au-assisted exfoliation and commonly used Scotch/Nitto tape method.
4. What is the success rate of obtaining flakes as large as ones shown in Fig. 2? It should be stated clearly in the paper.
5. The thickness of Au/Ti seems to be a crucial factor in the authors' experiment. The authors tried exfoliation on different thick Au/Ti as shown in Fig. 4b. How does the thickness of Au and Ti affect exfoliation productivity? What is the optimal thickness of Au and Ti? When the gold film is non-conductive and rough as shown in Fig 4b or with hole array as shown in Fig. 2j, does the exfoliation success rate drop due to the less contact interface between gold and van der Waals crystal?
6. It is well known that the reproducibility and productivity of exfoliation highly rely on lots of trivial details. The referee found the following information missing in the paper, which could be crucial for readers to reproduce the experiment. Where are the bulk crystals from? Which type of tape did the authors use? Does the thickness of bulk crystal on tape affect the exfoliation? What are the temperature and humidity of the lab? How long after exposure to air do the authors perform the Au-assisted exfoliation? Ref.23 has proved this factor to be crucial.

The minor ones are further listed below.

1. The novelty of the TMDC heterostructures lies in their interlayer excitons and Moiré excitons. It will strengthen the paper if the authors can show the interlayer excitons (~ 1.55 eV and/or 1.0 eV)

of the WSe₂/MoS₂ heterostructure fabricated in Fig. 2h.

2. Does the exfoliated monolayer black phosphorus in Fig. 2f show PL as reported?
3. The glue from the tape is expected to leave residues onto the silicon chip where there was no bulk crystal. How did the author avoid that?
4. In the caption of Fig. 2f-g, the authors claimed, "Those exfoliated monolayers highlighted in the red box are, so far, not accessible using other mechanical exfoliate method." The red box highlighted includes PtSe₂ and CrSiTe₃, which seem to be reported before (PtSe₂ in Nat. Commun. 9, 1545 (2018); CrSiTe₃ in J. Mater. Chem. C 4, 315 (2016)). The claim in Line 191 should be revised correspondingly.
5. Line 334, the authors better stick with using "gold-assisted" rather than introducing another word of "gold-enhanced".
6. What ARPES stands for is missing in the main text.
7. Figure S10 is not readable because the figure caption doesn't refer to four figures with labeling. It is not clear which ones are before annealing and which ones are after annealing. The description of the reason for this comparison is also missing.

Reviewer #2 (Remarks to the Author):

The authors report an impressive and well-documented technique to exfoliate large-area monolayers of a broad range of layered materials on substrates coated with gold. I think that in many (but definitely not all) cases, this technique will provide a significant boost to the community, particularly for those working on materials, such as black phosphorus or monochalcogenides, for which obtaining monolayers ranges from very difficult to impossible. In such cases, exfoliation on gold could become an enabling technique to permit progress where previously it was impossible. The obvious shortcoming of the method is that there is gold left behind underneath the flakes, which will be a limitation for the fabrication of the highest quality devices. Despite this shortcoming, I recommend publication of this work after the authors address the following points:

1. The manuscript provides insufficient discussion of the fact that gold is left behind at the end of the process, and I think this fact will somewhat limit the widespread adoption of the technique. The authors state in the abstract and elsewhere that the new method is "contamination free", but the presence of gold everywhere under the device certainly counts as a form of contamination. The authors take the approach to use very thin gold that is not electrically conducting, which is certainly helpful, but having an array of gold nanoparticles underneath a flake is very different from having a flake sitting on a uniformly insulating surface. A strong case is made that optical properties of the flakes are not strongly affected by the presence of the gold layer, but the case for electrical performance is weaker. The choice to consider an ionic liquid-gated MoS₂ device instead of a standard metal-dielectric device seems unusual, and leaves open the possibility that the performance of a more traditional metal-dielectric-gated device is degraded by the presence of the gold underneath the flake. The authors should include data for back-gated or top-gated devices and compare either Hall (preferably) or field-effect mobilities with devices based on flakes produced via regular tape exfoliation.
2. The exclusion of Pt from consideration is not well-motivated. The high melting point is not really a barrier since it can be deposited in most electron beam evaporators.
3. A heterostructure of MoS₂ and WSe₂ is reported, but there is no discussion of how it was fabricated. Was each layer separately exfoliated onto Au and one of them picked up, or was the top layer exfoliated directly onto the first layer? Another shortcoming of this approach is that it may make pick-up of large monolayers impossible because of the strong adhesion. Clarification about how the heterostructure was made could help address this criticism.

My remaining comments are in regard to the style of presentation and/or clarity of the manuscript:

4. The last sentence of the abstract mentions "a common rule that underpins a universal route for producing large-area monolayers" but this common rule is left unspecified. If the common rule is simply to exfoliate onto gold, this could be stated succinctly in place of the vague statement that currently concludes the abstract.

5. The last sentence of the first full paragraph on p. 4 refers to "previous attempts" at gold-mediated exfoliation. Since the reports successfully demonstrated exfoliation of monolayers, they should not be referred to as "attempts". Similarly, on p. 11, the authors state that "even some previous studies *claimed* to exfoliate large area chalcogenide layers," as if to cast doubt on the validity of the published work of others.

6. The description of how the procedure actually works (last full paragraph on p. 6) is unclear. In particular, it is not clear whether the bulk crystal itself is pressed onto the Au-covered substrate, or if tape is used to peel thick layers off of the crystal and then the tape is pressed onto the Au-covered substrate. The description of the procedure in the methods section is similarly unclear. Clarifying the precise procedure is crucial for a paper reporting a new exfoliation technique. A supplemental video would be extremely helpful.

Reviewer #3 (Remarks to the Author):

Let me first admit that the comments below have been based on a relatively superficial reading of the manuscript.

The authors use gold to overcome the interaction energy between layers in previously "difficult" 2D materials, which is an interesting idea.

My immediate impression is that the argument that the "gold doesn't do any big difference" is not that convincing. The doping of gold decorated TMDs is described in literature (<https://iopscience.iop.org/article/10.1088/2053-1583/1/3/034001>), and the strong binding between the TMD and the gold that the method is based on, should also matter. The authors say that themselves that there is covalency involved between the gold and the 2D materials, so it is a bit limiting that the crystals cannot easily come off and that they – at least for monolayers – could well be electronically affected by the strong bonding to Au surface. The capability to lift off crystals (by etching the gold, not very elegant) whilst preserving the high quality of the crystals would have made the approach more impactful.

It is a nice detail that the size of the monolayers is large enough to allow regular ARPES, and conducting enough to do STM/STS, but how to trust it?

It is also nice to see that the authors have transistor-like behavior in their Fig 4a, but that is achieved by electrolyte gating, not with a "normal" field effect. It is not clear to me whether it's possible at all to "gate it with a gate".

My reason against it is that it doesn't work as well as the authors would like us to believe, and that the novelty compared to <https://pubs.acs.org/doi/pdf/10.1021/acsnano.8b06101> is perhaps too low for publication in NC.

My reason I like it, is that the idea of using ultrathin metal layers that provide adhesion but are electrically discontinuous is new and potentially very useful, and that the shortcomings in this work may be ironed out in following papers.

I have not fully assessed the quality of the methodology, analysis etc, but in terms of concept, novelty and impact, I am slightly in favor of publishing, if the authors can show that the crystals can be used in an ordinary FET. Afterall, they are mentioning the potential for practical applications , and arguing that this may "ironically" bring exfoliation at par with silicon single crystal wafer production, so this is only fair. It should be straightforward to make FETs, and the method appears to be fairly straightforward as well (that is a selling point), so I don't think it is a lot to ask. If not, probably not worthy of Nature Communications.

best regards

Point-by-point reply to referees' comments and list of changes

Reviewer #1

Huang et al report a millimeter-scale 2D material exfoliation method assisted by the thin Au films deposited on the substrates. Au-assisted exfoliation was reported before to produce millimeter-scale transition-metal dichalcogenide (TMDC) monolayers. However, in this work, the authors have successfully extended the method to exfoliate up to 40 types of van der Waals materials, including semiconductors, magnets, and superconductors. It was not expected that Au-assisted exfoliation can work so well to different van der Waals materials other than TMDCs. The authors further demonstrated that their samples are suitable for angle-resolved photoemission spectroscopy (ARPES), scanning tunneling microscopy (STM), transport and optical measurements, which is not trivial since ARPES and STM are very sensitive to the sample quality and surface contaminations. These extensions certainly make the paper of interest to broader audiences. During the review process, the referee noticed another Au-assisted exfoliation paper published online on Science (Liu et al., Science 367, 903–906 (2020), online at 21.02), which demonstrates even larger exfoliated monolayers but still limited to TMDCs. The referee is convinced that this manuscript would be a very good complementary paper to the Science one. Thus, I would highly recommend the paper to be published on Nature Communications.

We thank the reviewer for his/her careful, constructive and positive review of our work. We appreciate for his/her recommendation of publication. We thus address all concerns raised by the reviewer as follows.

Nevertheless, there are still several issues to be addressed before the publication.

The major ones are:

Comments #1. *The charge density plot strongly depends on the electronic state chosen. What are the states used for charge density plots in Fig. 1c-f? How were these states chosen and are they related to the Fermi level?*

Reply1: We regret with our not-clear-enough presentation that confused the reviewer. The mentioned charge densities in Fig. 1c-f are differential charge densities (DCDs), which represent charge variation after a 2D layer attaching to the Au substrate and were derived with $\Delta\rho_{DCD} = \rho_{all} - \rho_{Au} - \rho_{2D}$. Here ρ_{All} is the total charge density of 2D layer/Au(111) interface, while ρ_{Au} and ρ_{2D} are total charge densities of an individual Au surface and a 2D layer, respectively. Light-coral and light-green isosurface represent charge accumulation and reduction, respectively. Such plots were used in many papers in the literature to show the electronic interaction between the adsorbates and substrates, e.g. DOI: 10.1038/s41598-019-48953-0, 10.1038/s41524-019-0161-8, 10.1039/c7cp01852e and 10.1103/PhysRevB.81.155403.

Action1: We updated the methods section by adding an explanation of DCD on page 14.

Comments #2. The data from STM and ARPES measurements in Fig. 3 looks very decent. But it is not clear how the samples were processed. What is the thickness of Au and Ti for the STM and ARPES samples? In general, monolayers for STM and ARPES characterization need to be annealed at a high temperature for hours to remove surface contaminations. Was the annealing process needed/performed for the authors' samples before the characterization? What was the size of the beam used for ARPES?

Reply2: We are grateful to these constructive comments. We used an Au(5 nm)/Ti(2 nm) substrate to exfoliate the samples for STM and ARPES measurements. Since the samples were exposed to air, a degas treatment is needed after transferring them into ultra-high vacuum (UHV). Therefore, we annealed the samples at 500 K for 2 hours in UHV before performing our STM and ARPES measurements. Such a procedure might be further optimized by exfoliating the samples in a glove box in future. The ARPES measurements for WSe₂ and MoS₂ monolayers were carried out at roughly 30 K using a home-build photoemission spectroscopy system with a VUV5000 Helium lamp. The spot diameter of the Helium lamp was 0.5 mm.

Action2: We added these details in the revision (Materials and Methods). ...“For STM and ARPES measurements, the 2D layers were exfoliated onto an Au(5nm)/Ti(2nm)/SiO₂(~300nm)/Si substrate. An annealing process at 500 K for 2 hours was performed to degas the samples after loading them into the high-vacuum chamber.”... “The ARPES measurements for WSe₂ and MoS₂ monolayers were carried out at roughly 30 K using a home-build photoemission spectroscopy system with a VUV5000 Helium lamp. The spot diameter of the Helium lamp is 0.5 mm.”

Comments #3. The authors claimed that the PL of their TMDC monolayers is not strongly quenched (Line 271) because of the thin layer of gold used. However, the PL spectrum of monolayer MoS₂ in Fig. S3b doesn't look very strong with the current signal to noise ratio. The scale bar for the PL imaging in Fig. S3d is missing. Figure S9b indicates that more than 10 times PL quenching occurs when there is Au underneath. To justify the claim, the authors should compare the monolayers produced by Au-assisted exfoliation and commonly used Scotch/Nitto tape method.

Reply3: We thank the reviewer for questioning on these details. The PL intensity of MoS₂ indeed decreased on the Au(1.5nm)/Ti(0.5nm)/SiO₂/Si thin film compared with that on bare SiO₂/Si substrates (Scotch/Nitto tape method). As suggested by the reviewer, we compared the PL spectra on both Au and SiO₂ substrates, as Fig. S4 shown below, the contrast fluorescence intensity decreased about 70%. We also double checked the scale bars for Fig. S3c and S3d. In addition, we regret with confusing the reviewer with the “PL quenching data” shown in Fig. S9b of the original manuscript. This is not a PL quenching, but a 10 times enhancement of suspended samples, if we use the samples using Scotch/Nitto tape method as reference.

Action3: A direct comparison of the PL spectra recorded on the Au and SiO₂ substrates was depicted in Fig. S4 in the revision. The scale bar for the Raman mapping is available in Fig. S3d. We slightly modified the caption of Fig. S12 (Fig S9 of the original manuscript) to stand out that the comparison aims to show the enhancement of suspended samples.

Fig. S4. Comparison of PL spectra between MoS₂ monolayers exfoliated on the SiO₂/Si substrate using the common tape method (red curve) and on the Au/Ti film using the present gold-assisted method (black curve), respectively.

Comments #4. What is the success rate of obtaining flakes as large as ones shown in Fig. 2? It should be stated clearly in the paper.

Reply4: The success rate is more than 99.5 % as long as the thickness of Au and Ti larger than 2 and 1 nm, respectively. We ever success in >200 exfoliation attempts of these materials prior to writing this manuscript. However, we also noticed that the yield ratio dramatically decreases if the Au and Ti layers are thinner than 0.5 and 0.5 nm, respectively; this is, most likely, a result that the metal film is too thin to attach on substrate firmly.

Action4: We clarified this issue by adding the above description/discussion on page 15 in the Materials and Methods section of the revision.

Comments #5. The thickness of Au/Ti seems to be a crucial factor in the authors' experiment. The authors tried exfoliation on different thick Au/Ti as shown in Fig. 4b. How does the thickness of Au and Ti affect exfoliation productivity? What is the optimal thickness of Au and Ti? When the gold film is non-conductive and rough as shown in Fig. 4b or with hole array as shown in Fig. 2j, does the exfoliation success rate drop due to the less contact interface between gold and van der Waals crystal?

Reply5: This is a good point. The thickness of Au/Ti is indeed a crucial factor. As mentioned in the reply to Comment #4, once the thickness is larger than Au(2nm)/Ti(1nm), the success rate is more than 99.5 %. However, the flake size decreases to hundreds of micrometers if the Au thickness reduces to 1 nm, as ascribed to less adhesion between the Au layer and 2D layers. If the thicknesses of both Au and Ti layers reduce to 0.5 nm, the exfoliation productivity drops to nearly zero because of the too weak adhesion between the metal layers and the SiO₂ substrate. In consideration of all successful cases, the optimal thickness is Au(2nm)/Ti(2 nm).

In light of this, the success rate nearly maintains but the size of flakes slightly reduces when the gold film is non-conductive and rough, i.e. Au(1nm)/Ti(1 nm). The decreased thickness (<0.5 nm) of Au (Ti) weakens the adhesion between Au and 2D layers (metal and SiO₂/Si), largely lowering the success rate of exfoliation. In terms of the substrate with hole array, the success rate for

exfoliating suspended MoS₂ is also more than 99% once the metal thickness is larger than Au(2nm)/Ti(1nm) and diameter of hole smaller than 5 μm. The success rate obviously decreases if the hole diameter is larger than 10 μm.

Action5: We have added this discussion in the revised manuscript (on page 16).

“The thickness of Au/Ti layer is a crucial factor. Once the thickness larger than 2 nm Au/ 1 nm Ti, the success rate is more than 99.5%. However, the flake size will decrease to few hundred micrometers when the metal layer thinner than 1 nm Au/ 1 nm Ti. It’s hard to get 2D flakes when metal thickness thinner than 0.5 nm Au/ 0.5 nm Ti. The optimal thickness of Au/Ti is 2 nm/ 2 nm. Therefore, when the gold film is non-conductive and rough, the success rate will be affected. In terms of the substrate with hole array, the success rate for exfoliating suspended MoS₂ is also more than 99% once the metal thickness is larger than Au(2nm)/Ti(1nm) and diameter of hole smaller than 5 μm. The success rate decreases obviously if the hole diameter larger than 10 μm.”

Comments #6. It is well known that the reproducibility and productivity of exfoliation highly rely on lots of trivial details. The referee found the following information missing in the paper, which could be crucial for readers to reproduce the experiment. Where are the bulk crystals from? Which type of tape did the authors use? Does the thickness of bulk crystal on tape affect the exfoliation? What are the temperature and humidity of the lab? How long after exposure to air do the authors perform the Au-assisted exfoliation? Ref.23 has proved this factor to be crucial.

Reply6: We greatly appreciate the reviewer with pointing out these important details. Our bulk crystals were mainly supplied from Prof. Youguo Shi’s group at the Institute of Physics, CAS and Prof. Yanfeng Guo’s group at ShanghaiTech University, Shanghai, China. They are experts and have decent reputation in the community of bulk crystal synthesis. We also tested some crystals (such as MoS₂, WSe₂) from commercial companies, like HQ Graphene and 2D Semiconductors, all these crystals can be exfoliated into large area monolayers, and the size of monolayers mainly depends on the lateral size of bulk crystals. We mainly used the Scotch/Nitto tape to exfoliate 2D materials, other materials can be also used to cleave bulk crystals, for example, PDMS.

There is a range for optimal thickness of the bulk crystal. The crystal should be not too thick, i.e. > 1 mm, otherwise it is difficult to ensure a good interface contact between crystal and substrate. It is also unnecessary to significantly cut the bulk crystal down to few or tens of layers on the tape, because such a thin crystal may easily fragment into small pieces after cleaving several times by tape. We usually use a crystal that was cleaved 2-3 times from a pristine bulk crystal.

The temperature and humidity is not very sensitive for the exfoliation, the authors and some of our collaborators tested this exfoliation method in different countries (US, China, Singapore and Korea) and in different seasons (the whole year from Spring to Winter), we could always get large area 2D flakes with high success rate. In this paper, we exfoliate most of the 2D materials in clean-room, and the temperature and humidity always keep at 25 °C and 45%. While the exfoliation can be done in a glove box without exposure to air, the whole process lasts 1~2 minutes.

Action6: We provided a short video and more exfoliation details in the Methods part.

Comments #7. The novelty of the TMDC heterostructures lies in their interlayer excitons and Moiré excitons. It will strengthen the paper if the authors can show the interlayer excitons (~1.55 eV and/or 1.0 eV) of the WSe₂/MoS₂ heterostructure fabricated in Fig. 2h.

Reply7: We thank the reviewer for this constructive suggestion. The interlayer excitons of MoS₂/WSe₂ were reported in Phys. Rev. Lett. **123**, 247402 (2019)), which positioned at 0.97 eV (infrared range) at room temperature and shifts to an energy of 1.02–1.05 eV at cryogenic temperatures. The measurement range of our Raman/PL spectrometer (Horiba HR800) is, however, between 535 nm to 1000 nm, so that we cannot observe the peak below 1.1 eV. We showed our data in the figure below (Fig. S6 of the revision). We noticed that the peak sitting around 1.6 eV is bit asymmetric, i.e. slightly stronger signal between 1.5 – 1.6 eV and a kink at 1.58 eV, which might be an indirect indication of the interlayer exciton peak at 1.55 eV. In the past two months, we have very limited access to our labs and no access to other labs that are equipped with more powerful spectrometers because of the outbreak of the novel coronavirus (COVID-19). While the main focus of this work lies in exfoliation, we would be happy and we should show more compelling data to clarify this issue in future.

Action7: we added the PL data of MoS₂/WSe₂ heterostructure in Fig. S6 of the revision and discussed our data as possible as we can.

Fig. S6 PL spectra of MoS₂/WSe₂ heterostructure.

Comments #8. Does the exfoliated monolayer black phosphorus in Fig.2f show PL as reported?

Reply8: Monolayer black phosphorus is easy to oxidize in air, which makes it difficult to get PL signal. We have also been working on optimization of the exfoliation process to protect air-sensitive materials, which is challenging and will be, most likely, reported in our future work. Here, only Raman spectra was used to check the quality of our exfoliated samples because it can provide information about the composition, structure and stability of materials. The Raman spectra of exfoliated monolayer black phosphorus are measured in a vacuum chamber. Our Raman data are consistent with those reported in the literature [*Nano Lett.* 16, 7761 (2016)]. However, as mentioned above, we cannot carry out any new PL measurement in the near future because of the

outbreak of the coronavirus (COVID-19).

Comments #9. *The glue from the tape is expected to leave residues onto the silicon chip where there was no bulk crystal. How did the author avoid that?*

Reply9: This is a good question. The glue residues mainly depend on the tapes we use. Either the white tape (3M scotch) or blue tape (Nitto) leaves some glue on substrate, which can be removed by further annealing if required by the subsequent measurement. To completely avoid this, we can use our home-made PDMS tape to replace these Scotch/Nitto tapes as holding media, which make the substrate much cleaner as reported in *Nanotechnology* 29 (2018) 265203.

Action9: we add two sentences discussing this issue in the methods section of the revision.

“The glue residues mainly depend on which type of tape used. For the white (3M Scotch) or blue (Nitto) tape, there are some glue left on the substrate, which can be removed by further annealing if required by the subsequent measurement. However, we can also use our home-made polydimethylsiloxane (PDMS) tape to replace the Scotch or Nitto tape as holding media, which make the substrate much cleaner⁵⁰.”

Comments #10. *In the caption of Fig. 2f-g, the authors claimed, “Those exfoliated monolayers highlighted in the red box are, so far, not accessible using other mechanical exfoliate method.” The red box highlighted includes PtSe₂ and CrSiTe₃, which seem to be reported before (PtSe₂ in *Nat. Commun.* 9, 1545 (2018); CrSiTe₃ in *J. Mater. Chem. C* 4, 315 (2016)). The claim in Line 191 should be revised correspondingly.*

Reply10 & Action10: We thank the reviewer for this correction. We have modified the figure and sentence in the revised manuscript.

Comments #11. *Line 334, the authors better stick with using “gold-assisted” rather than introducing another word of “gold-enhanced”.*

Reply11 & Action11: We appreciate with Reviewer’s carefully reading of our paper. We have unified our expression with “gold-assisted” in the revision.

Comments #12. *What ARPES stands for is missing in the main text.*

Reply12 & Action12: Thanks for your notice. We added the full name of ARPES in the main text, which can be seen on page 8, ...“Figs. S3 and S4 show Raman, photoluminescence (PL) and angle-resolved photoemission spectroscopy (ARPES) of a typical exfoliated MoS₂ monolayer.”

Comments #13. *Figure S10 is not readable because the figure caption doesn’t refer to four figures with labeling. It is not clear which ones are before annealing and which ones are after annealing. The description of the reason for this comparison is also missing.*

Reply13 & Action13: We are grateful to this detailed comment. We regret with this unclear figure caption. The AFM images shown in Fig. S13 of the revision (S10 of the original version) illustrate that the metals (Au/Ti) deposited on SiO₂/Si substrate is not a continuous film but some metal clusters with boundaries, which can become more pronounced after annealing. In light of this, the substrate surface is still not conductive even an ultrathin metal layer was deposited on it. We rewrote the caption of Fig. S13 (Fig. S10 of the original version) and explained the reason why

such comparison was provided.

Reviewer #2

The authors report an impressive and well-documented technique to exfoliate large-area monolayers of a broad range of layered materials on substrates coated with gold. I think that in many (but definitely not all) cases, this technique will provide a significant boost to the community, particularly for those working on materials, such as black phosphorus or monochalcogenides, for which obtaining monolayers ranges from very difficult to impossible. In such cases, exfoliation on gold could become an enabling technique to permit progress where previously it was impossible. The obvious shortcoming of the method is that there is gold left behind underneath the flakes, which will be a limitation for the fabrication of the highest quality devices. Despite this shortcoming, I recommend publication of this work after the authors address the following points:

We thank the reviewer for his/her careful, constructive and positive review of our work. We appreciate for his/her recommendation of publication. We thus address all concerns raised by the reviewer as follows.

Comments #1. *The manuscript provides insufficient discussion of the fact that gold is left behind at the end of the process, and I think this fact will somewhat limit the widespread adoption of the technique. The authors state in the abstract and elsewhere that the new method is "contamination free", but the presence of gold everywhere under the device certainly counts as a form of contamination. The authors take the approach to use very thin gold that is not electrically conducting, which is certainly helpful, but having an array of gold nanoparticles underneath a flake is very different from having a flake sitting on a uniformly insulating surface. A strong case is made that optical properties of the flakes are not strongly affected by the presence of the gold layer, but the case for electrical performance is weaker. The choice to consider an ionic liquid-gated MoS₂ device instead of a standard metal-dielectric device seems unusual, and leaves open the possibility that the performance of a more traditional metal-dielectric-gated device is degraded by the presence of the gold underneath the flake. The authors should include data for back-gated or top-gated devices and compare either Hall (preferably) or field-effect mobilities with devices based on flakes produced via regular tape exfoliation.*

Reply1: We appreciate the reviewer for this detailed concern. While the gold nanoparticles underneath might bring doping effects to the exfoliated 2D materials, they could be removed by selective Au etchant using KI/I₂ solution. The remaining 2D layer could thus be transferred onto any substrates for building ordinary FETs. Since the transfer method has been demonstrated in Ref. 22, we did not describe them with great details in the original version. The back-gate does not work for non-conductive Au exfoliated MoS₂ since those metals, even in discontinued clusters, offer substantial electric screening. There are two primary considerations that we use ionic liquid gating. It could tune carrier density of 2D materials in a larger range in comparison with regular back-gate and/or top-gate devices using high-k dielectric layers or h-BN, which may introduce new physics in 2D materials, e.g. superconductivity of MoS₂ presented in this work (Fig. S15). It is also very flexible that there is no extra fabrication procedure, e.g. deposition of Al₂O₃ or HfO₂ or transferring h-BN onto the 2D layer, in building ionic liquid-gated devices. Given the previously reported capacitance value of $C_i = 1.3\sim 2 \mu\text{F}/\text{cm}^2$ for the ionic liquid [*J. Phys. Chem. C*, 119, 39, 22297 (2015); *Phys. Chem. Chem. Phys.*, 15, 8983-9006 (2013)], we derived the

field-effect mobility of 22.1 to 32.7 cm^2/Vs for our ionic-liquid-gated MoS_2 device (Fig. 4c). The estimated mobility value is very close to that of our ordinary back-gated FET built with transferred MoS_2 monolayer on SiO_2/Si ($\sim 30 \text{ cm}^2/\text{Vs}$) and is also comparable with those values reported in the literature, i.e. tens of cm^2/Vs [*ACS Nano* 8, 5, 4074-4099 (2014)]. While we have no access to our fab labs since the outbreak of COVID-19, we had fabricated 10 ordinary back-gated FETs using transferred large-size MoS_2 monolayers on the SiO_2/Si substrate before the outbreak. Their electric characteristics were shown in Fig. S17 for comparison, as suggested by the reviewer. The averaged mobility of $\sim 30 \text{ cm}^2/\text{Vs}$ show that the mobility of MoS_2 transferred from the gold film to the SiO_2/Si substrate is similar to that directly exfoliated on the SiO_2/Si substrate by normal exfoliation method.

Action1: We discussed the role of gold nanoparticles for affecting device performance and mentioned the screening issue of back-gated devices in the presence of metal layers in the revision. We presented the measured results of ten MoS_2 FET devices built using transferred monolayers on the commonly used SiO_2/Si substrate in Fig. S17. This result is also consistent with Ref. 22 [*Advanced Materials* 28, 4053-4058 (2016)], they also found the quality of MoS_2 didn't degrade after transferring from gold film to other substrate. We added a plot showing temperature dependent I-V curve of a monolayer MoS_2 device gated by ionic liquid (DEME-TFSI) at 4 V in Fig. S15 where it shows a superconducting transition at $T_c=4.5 \text{ K}$.

Fig. S15. Temperature dependent I-V curve of a monolayer MoS_2 device gated by ionic liquid (DEME-TFSI) at 4 V. The measurement temperature ranges from 2 K to 150 K. The inset is a zoom-in I-V curve at temperature from 2 to 15 K, from which a superconducting transition is indicated at $T_c=4.5 \text{ K}$.

Fig. S17. Electrical measurements of monolayer MoS₂ back-gated FET devices. (a) Relationship between source-drain current I_{sd} and back gate voltage V_g for source-drain voltage V_{sd} ranging from 1 to 2 V. (b) Statistics of on-off ratio and mobility of ten monolayer MoS₂ back-gated FET devices. The monolayer MoS₂ flakes were transferred to Si wafer (with 300 nm SiO₂ layer) by etching gold film in KI/I₂ solution and DI water cleaning, as demonstrated in Fig. S5. Then, the FET devices were fabricated by patterning, metal deposition and lift-off procedures.

Comments #2. *The exclusion of Pt from consideration is not well-motivated. The high melting point is not really a barrier since it can be deposited in most electron beam evaporators.*

Reply2: We agree with the reviewer. Our calculation results indicate the binding between Pt and many 2D materials is rather strong and may significantly change their electronic properties. In addition, Au is mechanically softer than Pt which may improve the interfacial contact under the gentle pressure applied during the exfoliation process.

Action2: We revised the statement of Pt and mentioned those two reasons that we prefer to use Au in the revision.

Comments #3. *A heterostructure of MoS₂ and WSe₂ is reported, but there is no discussion of how it was fabricated. Was each layer separately exfoliated onto Au and one of them picked up, or was the top layer exfoliated directly onto the first layer? Another shortcoming of this approach is that it may make pick-up of large monolayers impossible because of the strong adhesion. Clarification about how the heterostructure was made could help address this criticism.*

Reply3: We agree with this suggestion. The preparation procedure of the heterostructure is as follows. Firstly, we exfoliated large area MoS₂ and WSe₂ monolayers on two separate Au/Ti/SiO₂/Si substrates, respectively (see, Fig. S5). Then, PMMA was spin-coated onto the MoS₂ monolayer and then the sample was put into KI/I₂ solution. After roughly 10 hours of etching, the gold film was removed and the PMMA film together with the MoS₂ flake detached from the SiO₂/Si substrate. In order to clean the ion residual, the PMMA film was washed three times using DI water. The next step lies in using the WSe₂ sample to pick up the PMMA/MoS₂ film from water. Since both MoS₂ and WSe₂ flakes are several millimeters in size, no special alignment is need in this step if the twisting angle between these two layers is not specified. Additional baking at ~ 100 °C ensures the contact between these two layers. Next, we employed the same procedure, i.e. etching in KI/I₂ solution and three times washing using DI water, to remove the Au film from

WSe₂. A new substrate, e.g. SiO₂/Si was used to pick up the hetero-bilayer from Di water. Finally, we removed the PMMA layer using acetone.

Fig. S5. Fabrication process of the MoS₂/WSe₂ heterostructure.

Action3: We added some detailed descriptions of the fabrication process in the methods section of the revision and drew a new schematic figure for the fabrication process of heterostructure, as shown in Fig. S5.

Comments #4. The last sentence of the abstract mentions "a common rule that underpins a universal route for producing large-area monolayers" but this common rule is left unspecified. If the common rule is simply to exfoliate onto gold, this could be state succinctly in place of the vague statement that currently concludes the abstract.

Reply4 & Action4: We fully agree with the reviewer and thank him/her for this suggestion. We have modified this sentence as ...`Enhanced adhesion between the crystals and the substrates enables such efficient exfoliation, for which we identify a gold-assisted exfoliation method that underpins a universal route for producing large-area monolayers and thus supports studies of fundamental properties and potential application of 2D materials.`

Comments #5. The last sentence of the first full paragraph on p. 4 refers to "previous attempts" at gold-mediated exfoliation. Since the reports successfully demonstrated exfoliation of monolayers, they should not be referred to as "attempts". Similarly, on p. 11, the authors state that "even some previous studies *claimed* to exfoliate large area chalcogenide layers," as if to cast doubt on the validity of the published work of others.

Reply5 & Action5: We appreciate with these comments and all suggests were well adopted. We replaced these words using "previous reports" and "proved", and double checked the whole manuscript to avoid any other expression that may bring ambiguity.

Comments #6. The description of how the procedure actually works (last full paragraph on p. 6) is unclear. In particular, it is not clear whether the bulk crystal itself is pressed onto the Au-covered substrate, or if tape is used to peel thick layers off of the crystal and then the tape is pressed onto the Au-covered substrate. The description of the procedure in the methods section is similarly unclear. Clarifying the precise procedure is crucial for a paper reporting a new exfoliation technique. A supplemental video would be extremely helpful.

Reply6 & Action6: We fully agree with the reviewer and indeed this comment was shared by Reviewer #1. We substantially revised the description of our methods with much more details

provided and also recorded a short video to show the exfoliation process. While we could, in principle, have a better video, the present one is the best we can do so far because of the very limited access to our labs due to the outbreak of the coronavirus (COVID-19).

Reviewer #3

Let me first admit that the comments below have been based on a relatively superficial reading of the manuscript.

The authors use gold to overcome the interaction energy between layers in previously “difficult” 2D materials, which is an interesting idea.

We thank the reviewer for his/her positive evaluation of our work and appreciate the time that the reviewer spent on this manuscript.

Comments #1. *My immediate impression is that the argument that the “gold doesn’t do any big difference” is not that convincing. The doping of gold decorated TMDs is described in literature (<https://iopscience.iop.org/article/10.1088/2053-1583/1/3/034001>), and the strong binding between the TMD and the gold that the method is based on, should also matter. The authors say that themselves that there is covalency involved between the gold and the 2D materials, so it is a bit limiting that the crystals cannot easily come off and that they – at least for monolayers – could well be electronically affected by the strong bonding to Au surface. The capability to lift off crystals (by etching the gold, not very elegant) whilst preserving the high quality of the crystals would have made the approach more impactful.*

Reply1: We thank the reviewer for these concerns which were well considered in the revision. We address the reviewers’ concerns as follows.

As suggested by the reviewer, in the revision, we showed the capability of lifting off those exfoliated 2D layers by etching the Au of the sample in KI/I₂ solution. Back-gated FETs were made and measured based on those layers, which show competitive mobility of 30 cm²/Vs and high on-off ratio of 10⁷ for monolayer MoS₂ FETs (Fig. S16 in the revision). We also emphasize that the effect of Au on those 2D layers could, almost, be eliminated in our free-standing samples (Fig. 2j, Fig.S11, and Fig.S12).

We agree with the reviewer that doping might be significant for Au decorated TMDs, e.g. WSe₂ as reported in the 2D Materials 2014 paper. The PL signal of the Au decorated WSe₂ was completely quenched in that paper (Fig. 2d and 2e), however, we could observe appreciable PL signal in our samples with a certain Au thickness, see Fig. S4; this indicates a substantial difference of our samples from the previously decorated samples. Our original statement was intended to stand out such difference.

We regret that our description of DFT calculations confuse the reviewer. Yes, we did find covalent-likely characteristic at, e.g. the Au-MoS₂ interface. Such characteristic, we call it quasi-bonding, is not a real chemical bonding, but a description of overlapped wave-functions between Au and S, which is driven by attraction arisen from the non-local correlation of their electrons (also known as dispersion force). We compared the electronic structure of MoS₂ of the MoS₂/Au interface (Fig. S1a) with that of an intact MoS₂ layer (S1b), which shows the shape of the original valence and conduction bands nearly maintained, especially around the band edges.

Action1: We developed the protocol for lifting off the crystals from the Au film (Fig. S5) and demonstrated its high quality using electric measurements in ten back-gated devices (Fig. S17) in our revision. We also provided more details and discussion in the revision.

Comments #2. It is a nice detail that the size of the monolayers is large enough to allow regular ARPES, and conducting enough to do STM/STS, but how to trust it?

Reply2: We thank the reviewer for his/her positive assessment of our STM/STS and ARPES results. The size and conductivity of our samples were verified by various techniques. During the review process of this work, an Au-assisted exfoliation paper was, as also mentioned by Reviewer #1, published in Science [Liu et al., *Science* 367, 903–906 (2020)], in which ARPES results were reported for TMDs exfoliated on Au film. We also noticed that STM images of exfoliated MoS₂ on thick Au films were shown in the literature [*Scientific Reports* 5, 14714 (2015)]. *Nano Research* 12, 3095–3100(2019), and our previous paper, explicitly reported the band structure of a large-size WSe₂ monolayer exfoliated using the Au-assisted method. These STM/ARPES results from different groups are consistent with those results acquired on samples prepared using other methods, e.g. tape exfoliation, CVD or MBE [*ACS Nano* 11, 12001–12007 (2017); *Nano Lett.* 14, 5, 2443-2447 (2014)]. In light of this, we have confidence that the STM/STS and ARPES results shown in this manuscript are not artefacts and this universal exfoliation method is thus reliable for STM/STS and ARPES characterizations.

Comments #3. It is also nice to see that the authors have transistor-like behavior in their Fig 4a, but that is achieved by electrolyte gating, not with a “normal” field effect. It is not clear to me whether it’s possible at all to “gate it with a gate”.

Reply3: We thank the reviewer’s comment. The exfoliated large area 2D layers could either be directly used to process top gate FETs or be transferred onto other substrates for building bottom/top gate devices. The back-gate does not work in the presence of metal films since they, even in discontinued clusters, offer substantial electric screening. Due to the outbreak of the novel coronavirus (COVID-19), we have no access to our fab labs, so that the demonstration of ordinary top-gated FETs is unlikely in the near future. Fortunately, we had fabricated 10 bottom-gated FETs based on transferred large-size MoS₂ monolayers on SiO₂/Si substrates before the outbreak (see reply to comment #1), which show high mobility (~30 cm²/V·s) and large on-off ratio (~10⁷), (Fig. S17 of the revision). Besides, as we replied for Reviewer 2 above, we derived the field-effect mobility of 22.1 to 32.7 cm²/Vs for our ionic-liquid-gated MoS₂ device (Fig. 4c). The estimated mobility value is very close to that of our ordinary back-gated FET built with transferred MoS₂ monolayer on SiO₂/Si (~30 cm²/Vs) and is also comparable with those values reported in the literature, i.e. tens of cm²/V.s [*ACS Nano* 8, 5, 4074-4099 (2014)].

Fig. S17. Electrical measurements of monolayer MoS₂ back-gated FET devices. (a) Relationship between source-drain current I_{sd} and back gate voltage V_g for source-drain voltage V_{sd} ranging from 1 to 2 V. (b) Statistics of on-off ratio and mobility of ten monolayer MoS₂ back-gated FET devices. The monolayer MoS₂ flakes were transferred to Si wafer (with 300 nm SiO₂ layer) by etching gold film in KI/I₂ solution and DI water cleaning, as demonstrated in Fig. S5. Then, the FET devices were fabricated by patterning, metal deposition, and lift-off procedures.

Action3: As mentioned in point 1, we added the back-gated MoS₂ FET data in Fig. S17 and provided more discussion on page 13 in the revision. It reads “For those FET devices with discontinuous metal layer underneath, the back gate does not work due to the screen effect. However, the 2D layers could be lifted off and transferred to another insulating substrate, e.g. oxidized Si wafer, by etching the gold layer. We also fabricated back-gated MoS₂ monolayer FET devices based on freshly transferred large-size MoS₂ flakes on SiO₂/Si substrates. Figure S17 presents the results of electrical transport measurements of the back-gated FETs, which show high mobility ($\sim 30 \text{ cm}^2/\text{V}\cdot\text{s}$) and large on-off ratio ($\sim 10^7$). These values manifest that the transferred MoS₂ monolayer still preserves its high quality (Fig. S17; see Methods for details).”

Comments #4. *My reason against it is that it doesn't work as well as the authors would like us to believe, and that the novelty compared to <https://pubs.acs.org/doi/pdf/10.1021/acsnano.8b06101> is perhaps too low for publication in NC. My reason i like it, is that the idea of using ultrathin metal layers that provide adhesion but are electrically discontinuous is new and potentially very useful, and that the shortcomings in this work may be ironed out in following papers.*

Reply4: We appreciate with the reviewer's commendation on the idea of using discontinuous metal adhesion layers. We were delighted to learn the reviewer's perspective that the shortcomings may be ironed out in following papers. We were also happy to share with the reviewer the novelty of this work acknowledged by Reviewers #1 and #2.

Comments #5. *I have not fully assessed the quality of the methodology, analysis etc, but in terms of concept, novelty and impact, I am slightly in favor of publishing, if the authors can show that the crystals can be used in an ordinary FET. After all, they are mentioning the potential for practical applications, and arguing that this may “ironically” bring exfoliation at par with silicon single crystal wafer production, so this is only fair. It should be straightforward to make FETs, and the method appears to be fairly straightforward as well (that is a selling point), so I don't think it is a lot to ask. If not, probably not worthy of Nature Communications.*

Reply5: Again, we thank the reviewer for his/her assessment of “in favor of publishing”. It was, compared with the CVD and MBE methods, quite slow that the development of mechanical exfoliation technique for preparing 2D monolayers. We agree with the reviewer that there is still a long way to go for exfoliation techniques. As we answered to point #3, we have fabricated ordinary back-gated FETs based on the exfoliated large-size monolayer crystals by transferring them onto insulating substrates (Fig. S17). The transferred large-size MoS₂ monolayer from gold film still show high quality and the measured mobility is similar to the values of monolayer MoS₂ exfoliated by common methods.

Reviewer #1 (Remarks to the Author):

The authors have made substantial changes to the manuscript and addressed all my questions and comments raised in the initial review. I highly appreciate the exfoliation video made by the author as suggested by the referee#2. Together with significant details now included, it will largely help readers to reproduce the experiment. The authors should refer to this video in the main text or method section.

I recommend the publication of the revised manuscript after addressing a minor issue below.

It is not clear why the baseline of MoS₂ PL spectra in the new Fig. S4 is not zero. The authors might want to correct it.

Reviewer #2 (Remarks to the Author):

I believe the authors have satisfactorily responded to my questions and criticisms, as well as those of the other referees, and I recommend publication of this manuscript in Nature Communications. In particular, the many details provided about the exfoliation process, heterostructure formation, etc. were very helpful, including the video that was provided.

Reviewer #3 (Remarks to the Author):

Im happy with the revision, and the paper can be published in NC